# Higher number of steps is related to lower endogenous progesterone but not estradiol levels in women

Kinga Słojewska[1,2], Andrzej Galbarczyk[1,3]*, Magdalena Klimek[1], Anna Tubek-Krokosz[1,2], Karolina Krzych-Miłkowska[1], Joanna Szklarczyk[4], Magdalena Mijas[1], Monika Ścibor[1], Grazyna Jasienska[1]

1 Department of Environmental Health, Faculty of Health Sciences, Jagiellonian University Medical College, Krakow, Poland, 2 Jagiellonian University Medical College, Doctoral School of Medical and Health Sciences, Krakow, Poland, 3 Department of Human Behavior, Ecology and Culture, Max Planck Institute for Evolutionary Anthropology, Leipzig, Germany, 4 Department of Medical Physiology, Faculty of Health Sciences, Jagiellonian University Medical College, Krakow, Poland

* agalbarczyk@gmail.com

## Abstract

### Objectives

Sex steroid hormones are important not only for reproduction but also for many aspects of women's health, including the risk of breast cancer. Physical activity has been shown to influence sex hormone levels in women. This study aimed to investigate a relationship between the average daily number of steps and the sex hormone (estradiol and progesterone) levels in premenopausal women.

### Materials and methods

Data were collected from 85 healthy, urban women of reproductive age who performed at least 180 minutes/week of moderate physical activity for two complete menstrual cycles. Physical activity was measured using wrist bands. Estradiol and progesterone concentrations were measured in daily-collected saliva samples in the second menstrual cycle.

### Results

There was a significant negative association between the average number of steps taken daily and salivary progesterone levels after adjusting for potential confounding factors (age, BMI). Women who took more than 10,000 steps a day had significantly lower progesterone levels compared to women who took less than 10,000 steps. The association between physical activity and estradiol levels was statistically insignificant.

### Discussion

Our results indicate that taking at least 10,000 steps a day reduces progesterone levels, but this intensity of physical activity may not be high enough to affect estradiol levels. Daily step tracking is a valuable element of health promotion, but currently recommended levels of

**Data Availability Statement:** All relevant data are within the manuscript and its Supporting Information files.

**Funding:** This study was supported by a grant from the National Science Centre (2017/25/B/NZ7/01509) awarded to GJ; a grant from Priority Research Area FutureSoc and qLife under the program "Excellence Initiative – Research University" at the Jagiellonian University in Krakow (U1C/P04/NO.02.07) awarded to GJ, and the Salus Publica Foundation. The funders had no role in study design, data collection and analysis, decision to publish, or preparation of the manuscript.

**Competing interests:** The authors have declared that no competing interests exist.

physical activity may not be high enough for healthy premenopausal women to significantly reduce both sex hormone levels and thus their risk of postmenopausal breast cancer.

## Introduction

Sex steroid hormones are important not only for reproduction but also for many aspects of women's health. Levels of these hormones are associated with the risk of cardiovascular diseases, osteoporosis, mood disorders and depression [1–7]. Importantly, lowering the levels of sex hormones has also a significant impact on reducing the risk of breast cancer in women [8–10].

Levels of sex hormones change in response to many factors, and it is well documented that high energy expenditure has a suppressive effect on the ovarian function. Many studies have shown a negative relationship between energy expenditure resulting from sports activities and ovarian function [11–14]. Energy expenditure during exercise often stimulates compensatory mechanisms such as weight loss or energy conservation, resulting in inhibition of reproductive function, including reduced sex steroid levels [15]. In many studies, participants who engaged in high-intensity physical activity experienced significant reduction in estrogens and progesterone levels, sometimes leading to amenorrhea [16, 17]. For this reason, menstrual disorders are common in women professionally involved in sports [18, 19].

However, not every type of exercise leads to easily detectable changes in regularity of cycles. Moderate-intensity exercise has been shown to reduce levels of ovarian steroid hormones without visible influence on a cycle length or occurrence of menstruation [20, 21]. Women with high leisure time physical activity had lower estradiol levels [22, 23] and in premenopausal women estradiol levels decreased with increasing physical activity [24]. Inhibition of reproductive function has been also described in studies on populations where high levels of energy expenditure result from occupational physical work, for example related to seasonal agricultural work [25]. Importantly, physical work can have an inhibitory effect on levels of progesterone, regardless of the good nutritional status and positive energy balance of women [25].

Relationship between physical activity and sex hormone levels has been a matter of interest, due to a role of these hormones in initiation and progression of breast cancer [26, 27]. Although many researchers have found a link between physical activity and hormone levels in postmenopausal women [26, 28–33], fewer studies have looked at estrogen and progesterone levels in premenopausal women who are not professional athletes [12, 25, 34–36]. Due to the variability of hormone levels during the menstrual cycle, hormonal studies in reproductive-age women are methodologically challenging to perform at least 7–8 days per cycle in the follicular and luteal phases should be sampled (depending on a hormone studied) to reliably characterize hormone levels in a particular cycle [37].

Most studies on physical activity and sex hormones relied on a self-reported, questionnaire data about levels of physical activity of women, without actual measurements of energy expenditure. There are several methods to measure energy expenditure, but most of them are either not possible to use outside the laboratory or are too expensive for studies with a larger number of participants (e.g., doubly labelled water). Therefore, commonly used devices such as activity trackers (smartwatches, pedometers) are often used to assess physical activity by calculating a number of steps that a person takes during a day [38, 39]. This method is also potentially of great practical value given that most individuals monitoring their physical activity use similar devices

Walking is the most basic form of human physical activity [40]. There is a statistically significant inverse relationship between daily step count and the all-cause mortality [41].

Moreover, there has been a strong evolutionary selection for increased walking, as measured by the number of daily steps. Human hunter-gatherer ancestors walked 10,000 to 18,000 steps a day in search of food [42]. Only recently, especially in high-income industrial environments, has this most basic form of human physical activity declined [40, 42].

While the information about number of steps taken is easy to obtain with the use of step counting devices it is not clear how many steps per day are needed for reproductive age women to reduce levels of sex hormones, and thus reduce the risk of breast cancer. In order to address this question, we investigated whether the average number of steps taken per day is related to the levels of estradiol and progesterone in 85 healthy, non-smoking women of reproductive age who performed moderate physical activity for 2 complete menstrual cycles. We hypothesize that a greater number of steps taken per day will be associated with lower levels of sex hormones. We have also explored whether the widely recommended minimum number of steps (10,000 steps a day) was sufficient to have a reducing effect on sex hormone levels and, thus, is beneficial for the potentially lowering the breast cancer risk.

## Materials and methods

### Study participants

The participants in this study were 85 urban women (mean age 27.1; SD 4.3) from the city of Krakow in Poland. Women were recruited through advertisements in local media, social media and promotional campaigns in the period from October 2019 to March 2020. Firstly, the telephone interview was conducted in order to evaluate if each participant met the inclusion criteria. The selection of women for the study was based on the following criteria: age between 20 and 35 years, self-assessment of regular menstrual cycles (not varying by more than 5 days in length between the cycles), no gynecological and endocrine disorders, no hormonal contraceptives or other hormonal drugs used for 6 months before recruitment, no pregnancy or lactation for 6 months before recruitment, and no smoking. The inclusion criterion was also the lack of health contraindications for undertaking physical exercise (based on the diagnosis of a medical doctor who was a member of the research team). The study was performed in accordance with the Declaration of Helsinki. The research protocol was approved by the Bioethics Committee of the Jagiellonian University (decision number: 1072.6120.47.2018). Written informed consent was obtained from all participants in the study.

### General questionnaire

Women who met the criteria were invited to a meeting. During the meeting, each participant was interviewed by a trained study assistant. The general questionnaire included questions about demographics, lifestyle, health status and reproductive health.

### Anthropometric measurements

During the meeting, measurements of body height and weight were performed by a trained member of the research team. Body weight, body fat percentage and muscle mass were measured by bioelectrical impedance using the TANITA scale (Tanita BC-545N).

### Physical activity

We measured participants' physical activity for 2 consecutive menstrual cycles. At the beginning of the observation, each woman received a fitness club pass which allowed for unlimited training sessions and was asked to perform no less than 180 minutes/week of moderate to vigorous physical activity of her choice. Participants received the Fitbit Alta HR wristband

accelerometers (Fitbit, Inc.; San Francisco, CA, USA), and they were asked to wear it 24 hours a day (except showers, baths or swimming). Fitbit devices accurately assess step count during basic physical activities such as walking and running [43, 44]. Therefore, we decided to use the measurement of the average daily number of steps as an indicator of the level of physical activity in the analyzes.

### Salivary estradiol and progesterone assay procedure

During the second menstrual cycle, morning saliva samples were collected individually by participants starting on the first day of menstruation and then every day throughout the entire menstrual cycle. Participants collected at least 4ml of saliva samples into plastic, sterile tubes through passive drool and stored them in home freezers in the temperature of about − 20˚C. Next, the samples were transported in portable freezers from the participants' homes to the laboratory. To detect the day of ovulation, participants performed tests based on detection of luteinizing hormone (LH) of 10 mIU/ml test sensitivity (Horien Medical, Solec Kujawski, Poland) from cycle day 9 to day 18 or until the test showed a positive result. Saliva samples were analyzed using commercially available hormonal immunoassays of HS ELISA kits by Demeditec (Demeditec Diagnostics GmbH, Kiel, Germany) for 17-α-hydroxy-progesterone (sensitivity: 1.1 pg/ml, standard range: 10–2 400 pg/ml) and 17-β-estradiol (sensitivity: 0.72 pg/ml, standard range: 1–100 pg/ml). Hormones were measured in samples from 9 selected days. The analyzes were based on the average hormone levels of these 9 days. Estradiol was measured 4 days before the ovulation, on the day of the ovulation and 4 days after the ovulation. Progesterone was measured on the day of ovulation and for the next 8 days. If none of the ovulation tests were positive, the estradiol level was determined for days -18 to -10 of the reverse cycle days, and the level of progesterone was determined for days -14 to -6 of the reverse cycle days. Each sample was analyzed in duplicates. The intra- and inter-assay CVs for progesterone were 8.65% and 8.53%, respectively. In case of estradiol intra- and inter-assay CVs were 9.63% and 10.13%.

### Statistical analyzes

Multiple generalized linear models were used to test the effects of exercise (expressed as the average number of steps per day) on estradiol and progesterone levels. Hormone levels were the dependent variables, while physical activity was an independent predictor. The effects of potential confounders such as age and BMI on estradiol and progesterone levels were examined in all models. Separate analyzes were performed with mean estradiol and progesterone levels as dependent variables. First, the number of steps was analyzed as a continuous variable. Secondly, because it is recommended that young adults take about 10,000 steps per day [45], we divided women into 2 groups based on whether they took more or less than 10,000 steps a day. In addition, we divided the women into tertiles based on the number of steps they took (low activity = 8,209 steps; medium activity = 10,450 steps; high activity = 12,274 steps). As the values of estradiol and progesterone had a positively skewed distribution, gamma models with a log-link function were used. We also rerun these models with either age and body fat percentage or age and muscle mass as potential confounding variables. The data were analyzed using the IBM SPSS Statistics 28.0.1.0 version for Mac OS package and StatSoft Statistica 13.1 PL statistical package.

## Results

Descriptive statistics for the study group are presented in Table 1. There was no statistically significant differences between the groups of women who took less than 10,000 steps and

**Table 1. Baseline characteristics of study participants, their physical activity and sex hormone levels and results of independent samples t-tests.**

| | All participants | | | Group | | Student's *t*-test | |
|---|---|---|---|---|---|---|---|
| | Mean (SD) | | | Less than 10,000 steps mean (SD) | More than 10,000 steps mean (SD) | t | p |
| | (N = 85) | Min | Max | (n = 39) | (n = 46) | | |
| Age (years) | 27.1 (4.3) | 20.4 | 35.4 | 26.7 (4.0) | 27.5 (4.6) | -0,66 | 0,51 |
| Cycle length (days)[a] | 29 (3.95) | 22 | 51 | 29.1 (4.67) | 28.9 (3.59) | 0,21 | 0,83 |
| Body height (cm) | 164.8 (5.5) | 146.7 | 183.8 | 166.0 (6.4) | 163.8 (4.3) | 1,83 | 0,07 |
| Body weight (kg) | 59.3 (8.34) | 44.7 | 90.3 | 60.4 (7.9) | 58.5 (9.2) | 1,00 | 0,32 |
| BMI (kg/m$^2$)[b] | 21.8 (2.85) | 16.3 | 32.1 | 21.9 (2.5) | 21.8 (3.1) | 0,22 | 0,83 |
| Average steps/day | 10319.3 (2703.6) | 3380.8 | 16011.0 | 7988.2 (1805.6) | 12295.8 (1479.2) | -12,00 | <0,001 |
| Estradiol (pg/ml) | 4,1 (1.7) | 2.1 | 11.6 | 4.0 (1.8) | 4.2 (1.7) | -0,58 | 0,56 |
| Progesterone (pg/ml) | 100.4 (62.6) | 15.2 | 303.9 | 113.0 (66.1) | 89.7 (58.1) | 1,76 | 0,08 |

[a] during participation in the study

[b] BMI, body mass index

more than 10,000 steps in mean age, cycle length, body height, body weight and BMI (Table 1).

There was a significant negative linear association between physical activity (i.e., average number of steps taken daily) and salivary progesterone levels (p = 0.014), after adjusting for potential confounding factors (e.g. age and BMI) (Table 2). The association between physical activity and estradiol levels was not statistically significant (p = 0.688). Age and BMI were not significantly related to levels of sex steroid hormones.

**Table 2. The association between physical activity and salivary estradiol (E2) and progesterone (P) levels.**

| | | Linear association | | | Two groups | | | Three groups | | |
|---|---|---|---|---|---|---|---|---|---|---|
| | | Average daily number of steps | | | more or less than 10,000 steps | | | tertiles of daily number of steps | | |
| N = 85 | | Exp(b) | 95%CI | p | Exp(b) | 95%CI | p | Exp(b) | 95%CI | p |
| **Estradiol (pg/ml)** | | | | | | | | | | |
| Daily number of steps | | 0.000008 | -0.00004; 0.00002 | 0.69 | | | | | | |
| ≥ 10,000 steps[a] | | | | | -0.02 | -0.10; 0.05 | 0.56 | | | |
| Tertiles [b] | 1st | | | | | | | 0.04 | -0.07; 0.15 | 0.47 |
| | 2nd | | | | | | | 0.01 | -0.09; 0.12 | 0.80 |
| Age (years) | | 0.01 | -0.01; 0.02 | 0.60 | 0.003 | -0.02; 0.02 | 0.70 | -0.01 | -0.03; 0.01 | 0.34 |
| BMI (kg/m$^2$) | | 0.01 | -0.02; 0.04 | 0.42 | 0.01 | -0.02; 0.04 | 0.42 | -0.02 | -0.05; 0.15 | 0.28 |
| **Progesterone (pg/ml)** | | | | | | | | | | |
| Daily number of steps | | **0.00007** | **-0.0001; -0.00002** | **0.01** | | | | | | |
| ≥ 10,000 steps[a] | | | | | **0.14** | **0.005; 0.27** | **0.04** | | | |
| Tertiles [b] | 1st | | | | | | | **0.24** | **0.05; 0.42** | **0.01** |
| | 2nd | | | | | | | -0.05 | -0.23; 0.13 | 0.58 |
| Age (years) | | 0.03 | -0.004; 0.06 | 0.08 | 0.02 | -0.01; 0.06 | 0.15 | 0.03 | -0.01; 0.07 | 0.20 |
| BMI (kg/m$^2$) | | -0.04 | -0.09; 0.005 | 0.08 | -0.04 | -0.09; 0.01 | 0.10 | -0.05 | -0.11; 0.01 | 0.11 |

Note: Boldface indicates statistical significance (p<0.05).

Abbreviation: Exp(b), exponentiated coefficient; CI, confidence interval.

[a] Reference level: Women who took less than 10,000 steps

[b] Reference level: 3rd tertile

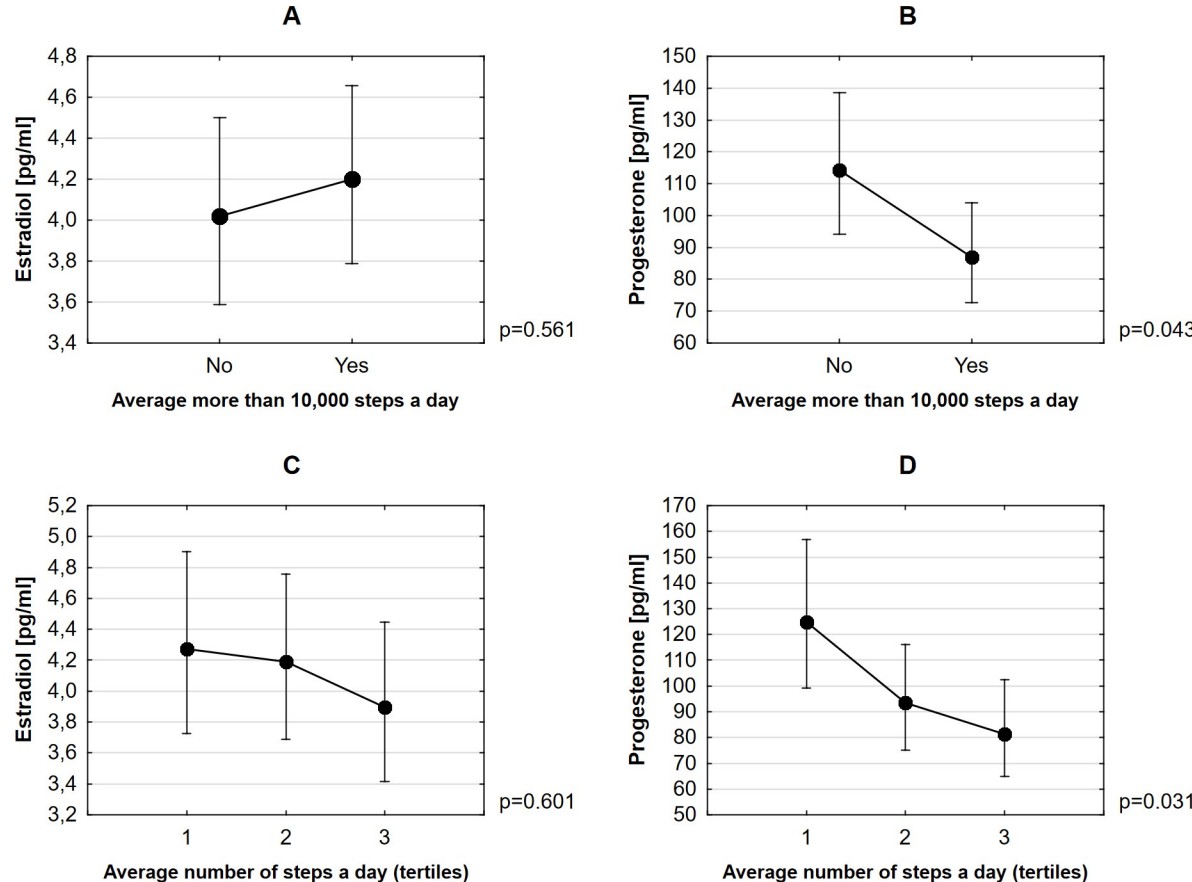

**Fig 1.** Differences between groups of women who took more or less than 10,000 steps in levels of estradiol (A) and progesterone (B), and between groups of women based on tertiles of the number of steps for estradiol (C) and progesterone (D). Lines and bars represent 95% confidence interval (CI) for predicted mean.

Women who took more than 10,000 steps a day had significantly lower progesterone levels compared to women who took less than 10,000 steps (p = 0.043), after adjusting for age and BMI (Table 2 and Fig 1). We found no statistically significant differences in estradiol levels in these activity groups. The model with division of the study sample into tertiles showed a statistically significant difference in the levels of progesterone among the examined women for the highest tertile compared to the lowest tertile, after controlling for age and BMI (p = 0.01). No statistically significant differences were observed for estradiol levels (Table 2 and Fig 1).

Next, we rerun our analyses replacing BMI with the body fat percentage (S1 Table) or muscle mass (S2 Table). There was a significant negative linear association between the average number of steps taken daily and salivary progesterone levels, after adjusting for body fat percentage and muscle mass. The model with division of the study sample into tertiles showed a statistically significant difference in the levels of progesterone among the examined women for the highest tertile compared to the lowest tertile, after controlling for body fat percentage and muscle mass. No statistically significant differences were observed in the estradiol levels (S1 and S2 Tables).

## Discussion

In this study we observed a significant, negative relationship between moderate physical activity, expressed as daily step count, and salivary progesterone levels. Women who took more

steps per day (thus were more physically active) had lower progesterone levels compared to women who took fewer steps. We did not observe statistically significant relationship between physical activity and estradiol levels.

The epidemiological studies show that physically active women have a lower risk of developing breast cancer [46, 47]. Not only sports and exercise, but also occupational work, domestic work and total physical activity are associated with a reduced risk of breast cancer [48]. The main physiological mechanism that is responsible for the relationship between physical activity and breast cancer risk is related to the effect of physical activity on ovarian function and its ability to reduce levels of ovarian steroid hormones [49–51].

It has been assumed that the main risk factor for breast cancer in women is exposure to high levels of estrogens, while the role of progesterone in the development of breast cancer is less clear. However, a recent review suggested that elevated levels of both hormones are in fact important risk factors [10]. According to a new hypothesis, progesterone stimulates proliferation of normal breast epithelium and also upregulates the expression of one of the DNA mutator enzymes. Estrogens, in turn, stimulate growth of already existing mutated, precancerous cells. Others suggested that progesterone may potentiate the mitogenic effects of estradiol [52] and animal models indicate that progesterone contributes to the proliferation of the endothelial gland and sensitizes cancer cells to growth factors [53–55]. In participants of our study only reduction of progesterone, but not estradiol, was observed in relation to physical activity. In the light of above described hypothesis, a reduction in progesterone could be sufficient to lower lifetime risk of breast cancer, even when physical activity does not change estradiol levels produced during menstrual cycles. The fact that we observe a response of ovarian function to physical activity in a form of lower progesterone levels, but not estradiol levels, is consistent with a classical model of ovarian suppression proposed by Ellison [56]. According to this model, ovarian function changes gradually in response to various stressors, especially those associated with energetics, such as physical activity. The various forms of ovarian suppression are not independent conditions but occur gradually. Between a fully fertile ovarian cycle and complete cessation of menstruation (i.e., amenorrhea), there are subtle degrees of variability in ovarian function. The mildest form of ovarian suppression consists of lowering progesterone secretion in the luteal phase of the cycle. The next stages may be inhibition of follicular development in the follicular phase and a lack of ovulation. A lack of menstrual bleeding is only the extreme end of a gradual continuum of ovarian responses to a changing environment [56].

In our analyzes, we decided to take the daily step count as the estimate of the level of physical activity, since walking is the most basic form of physical activity [40]. It is also an effective and simple way to increase level of physical activity. Taking 10,000 steps a day is equivalent to 30 minutes of moderate-to-vigorous physical activity [57–59] and this corresponds to the World Health Organization's recommendation of at least 150 minutes of moderate exercise per week [60]. In addition, daily step count is accessible information that people can use when they are concern about health recommendations and when they attempt to increase physical activity. While some women are aware that physical activity reduces the risk of breast cancer, they do not know if their own levels of activity are sufficient to make a change. Our results suggests that physical activity for premenopausal women, assessed by number of steps, should not be lower than 10,000 steps per day to reduce levels of progesterone, and thus, potentially reduce a risk of breast cancer.

It should be also emphasized that sex hormones, while related to the increased risk of breast cancer, are crucial for many aspects of healthy physiology [61, 62]. Their beneficial role is well established for fertility, cardiovascular function, bone health and mental health. However, insufficient levels of physical activity may lead to changes in the body's delicate hormonal balance contributing to altered hormone levels and adverse health outcomes. Thus, it is important for physically active women to discuss these issues with their physicians.

The strengths and limitations of our study should be taken into account. An important strength was that the hormonal analyzes were based on reliable assessment of hormone levels, as several saliva samples were analyzed for each participant, whereas still many studies rely only on a single or few measurements. Further, ovulation was determined based on LH ovulatory tests instead of a less reliable day-counting method. The limitation of our study could be the fact that number of steps per day was measured using the Fitbit wristband. Some previous studies have shown that Fitbit devices reliably estimate the number of steps and energy expenditure during basic physical activities such as walking and running [63]. Yet, this device gained a recent critique of not fully-reflecting the undertaken physical activity by an individual, both in laboratory and free-living settings (for a systematic review see: [44]). Future studies in this area involving a highly-accurate methods of physical activity measurement (e.g., doubly labelled water) are needed.

## Conclusions

In the quest to increase physical activity, daily step tracking is a valuable element of health promotion. In addition, it is one of the main and easily accessible functions supported by technology. Our results indicate that taking at least 10,000 steps a day may reduce levels of progesterone, but this intensity of physical activity may not be high enough to influence levels of estradiol. Potential health implications of exercise-induced changes in levels of sex hormones observed in this study are not well understood. However, currently recommended levels of physical activity may not be high enough to reduce levels of both sex hormones involved in development and progression of breast cancer and thus may not be sufficiently high for premenopausal healthy women to significantly reduce their risk of postmenopausal breast cancer.

## Supporting information

**S1 Table. The association between physical activity and salivary estradiol (E2) and progesterone (P) levels after controlling for age and body fat %.**
(PDF)

**S2 Table. The association between physical activity and salivary estradiol (E2) and progesterone (P) levels after controlling for age and muscle mass.**
(PDF)

**S1 Dataset.**
(XLSX)

## Acknowledgments

We are grateful to the women who participated in this study. This work could not have been completed without their involvement. We also thank all the study assistants who have helped with data collection.

## Author Contributions

**Conceptualization:** Kinga Słojewska, Andrzej Galbarczyk, Magdalena Klimek, Grazyna Jasienska.

**Data curation:** Kinga Słojewska, Magdalena Klimek, Anna Tubek-Krokosz, Magdalena Mijas.

**Formal analysis:** Kinga Słojewska, Andrzej Galbarczyk, Joanna Szklarczyk, Magdalena Mijas.

**Funding acquisition:** Grazyna Jasienska.

**Investigation:** Kinga Słojewska, Andrzej Galbarczyk, Magdalena Klimek, Anna Tubek-Kro-kosz, Karolina Krzych-Miłkowska, Monika Ścibor.

**Methodology:** Grazyna Jasienska.

**Project administration:** Magdalena Klimek, Anna Tubek-Krokosz, Karolina Krzych-Miłkowska.

**Supervision:** Monika Ścibor, Grazyna Jasienska.

**Writing – original draft:** Kinga Słojewska, Grazyna Jasienska.

**Writing – review & editing:** Andrzej Galbarczyk, Magdalena Klimek, Anna Tubek-Krokosz, Karolina Krzych-Miłkowska, Joanna Szklarczyk, Magdalena Mijas, Monika Ścibor.

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
