## [Decision Letter · Decision Letter 0]

28 Nov 2023

PONE-D-23-31898Higher number of steps is related to lower endogenous progesterone but not estradiol levels in womenPLOS ONE

Dear Dr. Galbarczyk,

Thank you for submitting your manuscript to PLOS ONE. After careful consideration, we feel that it has merit but does not fully meet PLOS ONE’s publication criteria as it currently stands. Therefore, we invite you to submit a revised version of the manuscript that addresses the points raised during the review process.

We look forward to receiving your revised manuscript.

Kind regards,

Sandar Tin Tin

Academic Editor

PLOS ONE

 [This study was supported by the National Science Centre (2017/25/B/NZ7/01509); Priority Research Area FutureSoc and qLife under the program “Excellence Initiative – Research University” at the Jagiellonian University in Krakow (U1C/P04/NO/02.07) and the Salus Publica Foundation.].  

4. Please include the reference section of your manuscript.

Reviewers' comments:

Reviewer's Responses to Questions

**Comments to the Author**

1. Is the manuscript technically sound, and do the data support the conclusions?

Reviewer #1: Yes

Reviewer #2: Partly

2. Has the statistical analysis been performed appropriately and rigorously? 

Reviewer #1: Yes

Reviewer #2: No

3. Have the authors made all data underlying the findings in their manuscript fully available?

Reviewer #1: No

Reviewer #2: Yes

4. Is the manuscript presented in an intelligible fashion and written in standard English?

Reviewer #1: Yes

Reviewer #2: Yes

5. Review Comments to the Author

Reviewer #1: In the study entitled "Higher number of steps is related to lower endogenous progesterone but not estradiol levels in women," the authors investigate the association between physical activity and the levels of sex hormones in premenopausal women. The importance of the study is driven by the paucity of data on the mechanism by which physical activity reduces the risk of breast cancer. The paper is well-written, the methodology and design of the study are clear, but there should be clarification regarding other physical activities the participants might have done during the study period.

Comments that should be acknowledged:

• Although mentioned in the limitation, please clarify if you asked about other physical activities unrelated to steps. For example, the Fitbit Alta HR wristband accelerometers cannot track biking in the step count. If women performed other activities not monitored by the device, it could significantly change the results

• Table 1 should be in the results section

• Statistical analysis: “multiple regression models were used to test the effects of exercise (expressed as number of steps) on estradiol and progesterone levels.” I would consider changing it to “… (expressed as the average number of steps per day)...”

• Table 2 – linear association between daily number of steps and estradiol level: I think the p-value should be 0.688 or 0.69 and not 0.60.

• The reference to Feehan et al on page 19 should be changed to [49].

Reviewer #2: The manuscript studied the relationship between female sex hormone levels and the level of physical activity in women. The topic is of high importance since the level of physical activity is very low in adults in the modern societies, and the prevalence of sex-hormone related cancers is increasing in women. The aims are well defined, however the methods that is used to estimate the influence of physical activity on sex hormone levels is not appropriate. The aims are well defined, however the methods are not appropriate.

I suggest the manuscript for publication after a major revision. The reasons for this suggestion are the following:

1) Two subgroups of women were formed on the basis of the level of physical activity, however, the two subgroups are not described by age and the other studied biological parameters (height, weight, BMI, the description is given only for the whole sample (Table 1). Since the age-group is ranged between 20.4 and 35.4 years, this is important to describe. Sex hormone levels change by ageing, a decrease can be seen after 30 years of age in the level of estradiol and progesterone. The mean steps per day is 10319 in the whole sample, this value is rather high in this age-group, it should be important to see the mean value of steps per day also for the subgroups. The recommendation for the studied age group for steps per day is 7-10000 according to the literature (e.g. DOI: https://doi.org/10.1016/S2468-2667(21)00302-9). By considering the mean steps of the 2 subgroups could help in the understanding of the research results. It is my impression that the cut-off value (10 000 steps/day) too high in the studied sample, or not only two but 3 subgroups should be formed (low, average, high level of physical activity) to analyse the relationship between the level of physical activity and sex hormone levels.

2) It is not clear why it is of high importance to ‘reduce’ the sex hormone levels in premenopausal women (this is emphasized by the Authors in the Abstract and in the main part of the manuscript, too). They are in the reproductive stage of their lives, decreased sex hormone levels can influence menstrual cycle, reproductive activity, bone health etc. It must be emphasized that the level of physical activity decreased very dramatically in the last decades, so increased level of physical activity means in this case that women have sufficient level of physical activity.

3) Why only body weight measured by Tanita was analysed in the manuscript? This device estimate body mass components as well, why the relationship of sex hormone levels with body fat mass and muscle mass were not analysed? Increased body fatness is evidenced to be related with sex hormone levels in women.

6. PLOS authors have the option to publish the peer review history of their article (what does this mean?). If published, this will include your full peer review and any attached files.

Reviewer #1: No

Reviewer #2: **Yes: **Annamaria Zsakai

---

## [Author Response · Author response to Decision Letter 0]

11 Jan 2024

Dear Editors,

We would like to thank the Reviewers and Editors for their insightful assessment of the manuscript. We have tried to adequately answer all reviewers’ comments. We believe that our work has improved significantly, and we hope our manuscript will be considered suitable for publication. 

Reviewer #1: 

Although mentioned in the limitation, please clarify if you asked about other physical activities unrelated to steps. For example, the Fitbit Alta HR wristband accelerometers cannot track biking in the step count. If women performed other activities not monitored by the device, it could significantly change the results.

-Response: We asked participants about other activities but decided not to take them into account in this publication. Due to the fact that there are official recommendations on how many steps a day should be taken to maintain health, we have decided to include only the average number of steps taken per day in this publication. Information about other activities was self-reported by participants and thus not very reliable and difficult to quantify. In addition, the purpose of this study was to assess if number of steps per day correlates with levels of ovarian hormones. This approach allows to formulate conclusions that are useful for individual women and important in the context of public health in the area of breast cancer prevention.

Table 1 should be in the results section

-Response: Thank you. Agreed. We moved Table 1 to the results section.

Statistical analysis: “multiple regression models were used to test the effects of exercise (expressed as number of steps) on estradiol and progesterone levels.” I would consider changing it to “… (expressed as the average number of steps per day)”

-Response: Thank you for pointing out this shortcut. As suggested, we changed "expressed as number of steps" to "expressed as average number of steps per day".

Table 2 – linear association between daily number of steps and estradiol level: I think the p-value should be 0.688 or 0.69 and not 0.60.

-Response: Thank you very much for spotting this mistake. We changed the p-value to 0.69.

The reference to Feehan et al on page 19 should be changed to [49].

-Response: We are not entirely sure if our interpretation of the Reviewer’s suggestion is correct. We have found an incorrect style of the reference “Feehan et al., 2018” on page 12. We have now changed it to the Vancouver style. We would be grateful if the Reviewer could confirm if this was the suggested issue.

Reviewer #2

Two subgroups of women were formed on the basis of the level of physical activity, however, the two subgroups are not described by age and the other studied biological parameters (height, weight, BMI) the description is given only for the whole sample (Table 1). Since the age-group is ranged between 20.4 and 35.4 years, this is important to describe. Sex hormone levels change by ageing, a decrease can be seen after 30 years of age in the level of estradiol and progesterone. The mean steps per day is 10319 in the whole sample, this value is rather high in this age-group, it should be important to see the mean value of steps per day also for the subgroups. The recommendation for the studied age group for steps per day is 7-10000 according to the literature (e.g. DOI: https://doi.org/10.1016/S2468-2667(21)00302-9). By considering the mean steps of the 2 subgroups could help in the understanding of the research results.

-Response: Thank you for pointing this out. We have supplemented Table 1 with descriptive characteristics of the subgroups. 

We agree, that the mean steps per day is rather high in this age-group. This is a result of the study's design. At the beginning of the observation, each woman received a fitness club pass which allowed for unlimited training sessions and was asked to perform no less than 180 minutes/week of moderate to vigorous physical activity of her choice. We have clarified this in the methods section. 

It is my impression that the cut-off value (10 000 steps/day) too high in the studied sample, or not only two but 3 subgroups should be formed (low, average, high level of physical activity) to analyse the relationship between the level of physical activity and sex hormone levels.

Response: We decided to set a cut-off of 10,000 steps a day due to the widespread recommendations of this value as necessary to maintain health. However, we realize that there is evidence in the literature that health benefits are observed at rates lower than 10,000 steps per day. Therefore, we also divided the study group into 3 subgroups (tertiles: low = 8,209; medium = 10,450; high = 12,274) according to the average daily number of steps. These results are presented in Table 2.

It is not clear why it is of high importance to ‘reduce’ the sex hormone levels in premenopausal women (this is emphasized by the Authors in the Abstract and in the main part of the manuscript, too). They are in the reproductive stage of their lives, decreased sex hormone levels can influence menstrual cycle, reproductive activity, bone health etc. It must be emphasized that the level of physical activity decreased very dramatically in the last decades, so increased level of physical activity means in this case that women have sufficient level of physical activity.

-Response: Indeed, this issue was not sufficiently explained in our original manuscript. We have added the following in the discussion:

It should be also emphasized that sex hormones, while related to the increased risk of breast cancer, are crucial for many aspects of healthy physiology [63,64]. Their beneficial role is well established for fertility, cardiovascular function, bone health and mental health. However, insufficient levels of physical activity may lead to changes in the body’s delicate hormonal balance contributing to altered hormone levels and adverse health outcomes. Thus, it is important for physically active women to discuss these issues with their physicians.

Why only body weight measured by Tanita was analysed in the manuscript? This device estimate body mass components as well, why the relationship of sex hormone levels with body fat mass and muscle mass were not analysed? Increased body fatness is evidenced to be related with sex hormone levels in women.

-Response: Although the role of body mass index on sex hormone levels and the development of breast cancer in premenopausal women is not clear, body mass index has been shown to be associated with sex hormone binding [Ingram, D M et al. 1990; doi:10.1038/bjc.1990.57]. Thus, we decided to perform statistical analyzes using BMI because it is a parameter that anyone can easily calculate. We wanted our research to be understandable to people outside the scientific community and to be easily conveyed in conversation. This is important in the context of public health. Additionally, in response to the Reviewer’s comment we rerun our analyses replacing BMI with the percentage of body fat or muscle mass.

---

## [Decision Letter · Decision Letter 1]

6 Feb 2024

PONE-D-23-31898R1Higher number of steps is related to lower endogenous progesterone but not estradiol levels in womenPLOS ONE

Dear Dr. Galbarczyk,

Thank you for submitting your manuscript to PLOS ONE. After careful consideration, we feel that it has merit but does not fully meet PLOS ONE’s publication criteria as it currently stands. Therefore, we invite you to submit a revised version of the manuscript that addresses the points raised during the review process.

We look forward to receiving your revised manuscript.

Kind regards,

Sandar Tin Tin

Academic Editor

PLOS ONE

Journal Requirements:

Reviewers' comments:

Reviewer's Responses to Questions

**Comments to the Author**

1. If the authors have adequately addressed your comments raised in a previous round of review and you feel that this manuscript is now acceptable for publication, you may indicate that here to bypass the “Comments to the Author” section, enter your conflict of interest statement in the “Confidential to Editor” section, and submit your "Accept" recommendation.

Reviewer #1: All comments have been addressed

2. Is the manuscript technically sound, and do the data support the conclusions?

Reviewer #1: Yes

3. Has the statistical analysis been performed appropriately and rigorously? 

Reviewer #1: Yes

4. Have the authors made all data underlying the findings in their manuscript fully available?

Reviewer #1: Yes

5. Is the manuscript presented in an intelligible fashion and written in standard English?

Reviewer #1: Yes

6. Review Comments to the Author

Reviewer #1: Thank you for revising the manuscript based on the Reviewers comments. I have one more comments that should be addressed: Please describe in table 1 if there is a significant difference in any of the parameters between the groups (<10000 steps per day and > 10 days per day). It is clear there is a difference in the number of steps, but other parameters should be compared.

7. PLOS authors have the option to publish the peer review history of their article (what does this mean?). If published, this will include your full peer review and any attached files.

Reviewer #1: No

---

## [Author Response · Author response to Decision Letter 1]

8 Feb 2024

As suggested by the Reviewer #1, we have added results of independent samples t-tests to the Table 1. We have also stated in the Results section that: "There was no statistically significant differences between the groups of women who took less than 10,000 steps and more than 10,000 steps in mean age, cycle length, body height, body weight and BMI."

We also made sure that we did not cite papers that have been retracted.

---

## [Editor Report · Decision Letter 2]

13 Feb 2024

Higher number of steps is related to lower endogenous progesterone but not estradiol levels in women

PONE-D-23-31898R2

Dear Dr. Galbarczyk,

We’re pleased to inform you that your manuscript has been judged scientifically suitable for publication and will be formally accepted for publication once it meets all outstanding technical requirements.

Kind regards,

Sandar Tin Tin

Academic Editor

PLOS ONE

---

## [Editor Report · Acceptance letter]

26 Mar 2024

PONE-D-23-31898R2 

PLOS ONE

Dear Dr. Galbarczyk, 

I'm pleased to inform you that your manuscript has been deemed suitable for publication in PLOS ONE. Congratulations! Your manuscript is now being handed over to our production team.

Kind regards, 

on behalf of

Dr. Sandar Tin Tin 

Academic Editor

PLOS ONE